# Stigma and Emotion Regulation in Intimate Partner Violence: A Pilot Exploratory Study with Victims, Offenders and Experts

**DOI:** 10.3390/bs15091229

**Published:** 2025-09-10

**Authors:** Christian Moro, Michela Scaccia, Teresa Camellini, Livia Lugeri, Emanuele Marrocu, Gian Piero Turchi

**Affiliations:** Department of Philosophy, Sociology, Education and Applied Psychology, University of Padova, 35131 Padova, Italy; christian.moro@unipd.it (C.M.); michela.scaccia@unipd.it (M.S.); teresa.camellini@unipd.it (T.C.); livia.lugeri@studenti.unipd.it (L.L.); emanuele.marrocu@studenti.unipd.it (E.M.)

**Keywords:** intimate partner violence, gender-based violence, stigma, emotion regulation, narratives, MADIT, clinical intervention

## Abstract

In the field of gender-based violence research, the social constructionist strand focuses on how stereotypes and discourses impact the psychological, socio-economical and sanitary levels of actors involved. Narratives of victims of intimate partner violence (IPV) often revolve around feelings such as shame, guilt and fear; these are related to emotion regulation. Considering this, narratives on how the roles of victims, but also offenders, are shaped are pivotal for clinical interventions. Considering gender-based violence as a product of human discursive interaction, this exploratory work investigates the terms used to describe these two roles and the ways in which those terms are conveyed. Ad hoc open-ended questionnaires were administered to 35 respondents from an Italian anti-violence centre, including IPV victims and offenders and healthcare experts. Their answers were analysed through MADIT (Methodology for the Analysis of Computerised Text Data), while the software IRaMuTeQ (version 0.7 alpha 2) was used for content analysis. Starting from the research question of “how do victims, offenders and experts groups narrate the roles of victim and offender”, the research hypothesis states that all three groups will adopt modalities that define the two roles into fixed and typical emotional categories. As anticipated, the results show that victims, offenders and experts depict both roles as immutable, categorising and judging the victims with words related with fear and self-guilt, while offenders are described with words related to anger and pathology. Lastly, we propose a framework for clinical intervention focused on fostering change towards a broader narrative to reduce the psychological impact of IPV events for victims, as well as modifying offenders’ violent behaviours.

## 1. Introduction

### 1.1. The Normative Framework of Gender-Based Violence and Intimate Partner Violence

The phenomenon of gender-based violence (GBV) and intimate partner violence (IPV) has become an increasingly sensitive topic in Italy in recent years; this is also because of crime-related news such as the femicide of Giulia Cecchettin (which occurred in the Veneto Region) by her ex-boyfriend ([40]), which led to numerous protests across the country. The attention paid to this event demonstrates the situation faced by the community in everyday life. According to the data collected by the ISTAT (Italian National Institute of Statistics), 31.5% of women aged between 16 and 70 experienced physical or sexual victimisation; in 62% of these cases, the violence was performed by a partner or ex-partner ([30]). Data from the anti-violence and stalking helpline highlight that 72% of the violent events reported (15,349 in the third quarter of 2024) occurred at home; this percentage remains unchanged compared to previous reports ([32]). Also, regarding cases of murder in 2023, 12% were labelled as a ‘crime of passion’ ([31]), while in [13] ([13]), it was observed that the victim’s partner was the offender in the 68% of episodes of domestic violence.

At the international level, the European Parliament and Council Directive define gender-based violence as direct violence against a person because of their gender, gender identity or gender expression or violence that affects people of a particular gender in a disproportionate way ([19]). Within the same Directive and the Convention of Istanbul ([15]), gender-based violence is defined as a discrimination act and a violation of the fundamental liberties of the woman in her role as victim and can cause physical, sexual, emotional, psychological or economic damage to the victim. Furthermore, a specific form of gender-based violence named IPV is defined by the WHO as a behaviour within an intimate relationship that causes physical, sexual or psychological harm, including acts of physical aggression, sexual coercion, psychological abuse and controlling behaviours, and this definition covers violence by both current and former spouses and partners ([75]).

European and Italian political–administrative apparatuses have developed materials to intervene with the actors involved in gender-based violence. Also, the scientific literature offers useful elements for the construction of dedicated interventions targeted towards the two main roles involved in IPV situations, i.e., victims and offenders. An example comes from Kelly and Johnson’s study, which emphasises the need to differentiate between types of IPV to improve the quality of screening programmes, decision making, sanctions and treatment ([36]). These materials led to the drafting of guidelines that can be used by policy makers and professionals to design and deploy their interventions on gender-based violence, specifically IPV.

### 1.2. The Scientific Literature on Gender-Based Violence and Intimate Partner Violence

Scientific research on GBV and IPV aims at offering an overview of the specific features of violence situations, focusing on the role of the victim and the possible repercussions of trauma, such as Post-Traumatic Stress Disorder and Depression Syndrome. For this reason, institutions provide many services, including emergency hotlines and programmes. An example is an intervention programme titled Supporting Survivors and Self (SSS) ([20]), which trains informal supports to increase victim empathy and reduce victim blame. In addition, it emphasises the importance of acknowledging survivors’ emotions and providing a safe space to process difficult memories. This approach can help victimised participants adopt coping strategies and seek out positive social support, thereby reducing the likelihood of developing symptoms of depression and PTSD.

Another line of scientific research aims at identifying the factors that can help in preventing and contrasting this phenomenon. Some of the risk factors regarding women involved as victims are income, education level ([29]; [54]) and age ([65]); for men involved as offenders, risk factors include narcissism ([45]), fragile masculinity, the sense of the right to sex ([64]) and personality disorders ([71]). Due to the intrapsychic nature of the risk factors for male perpetrators of GBV and IPV, integrated approaches have been designed to include cognitive-behavioural and pharmacological therapy to control sexual desire ([43]), together with the support of a sex therapist ([18]). Costa et al.’s study captured child abuse, family of origin, child and adolescent behaviour problems, adolescent substance use, adolescent peer risks and, less consistently, sociodemographic risk as significant predictors of both domestic violence perpetration and victimisation for men and women ([14]); many of these predictors are also risk factors for mental disorders ([51]). Baños et al. deepened the topic of intrapsychic aspects regarding anxiety and depressive disorders—which are more frequent in women—and disorders related to the abuse of psychoactive substances or antisocial personality—which are more frequent in men ([4]). Additionally, Exposito et al. studied the role of self-regulatory failure as a predictor of violent behaviour toward a partner ([22]).

Another intrapsychic risk factor associated with IPV in the scientific literature is emotion regulation, defined by Gratz & Roemer as “the ability to recognise, label, comprehend, express, accept, and manage emotions in the service of both short and long-term goals” ([26]). In fact, emotion regulation strategies have been shown to be malleable, and it is important for researchers to study these strategies when analysing the risk factors of IPV ([56]), both in cases of victims and offenders. Considering victims, one study ([27]) showed how cognitive reappraisal and expressive suppression, two types of emotion regulation strategies, mediated the effect of IPV victimisation on health. The authors found that at higher levels of habitual expressive suppression, IPV victimisation was associated with greater substance use and that at higher levels of habitual cognitive reappraisal, IPV victimisation was associated with worse general health and greater substance use. Other studies showed how emotion dysregulation acts as a causal mechanism and maintaining factor across a variety of problems and disorders, including those that often co-occur in survivors of IPV (e.g., PTSD, anxiety, depression, substance abuse and low self-esteem), and how it can lead people to develop maladaptive coping behaviours, including avoidance and other risky conducts ([58]). Considering offenders, recent studies showed how sadistic traits were related to several dimensions of emotion dysregulation, such as emotional nonacceptance, lack of emotional awareness, and poor distress tolerance ([72]). Regarding offenders, [62] ([62]) noted that anger and emotion dysregulation are correlated with emotional IPV perpetration. Furthermore, studies observed how aspects of emotion dysregulation—such as difficulties in refraining from impulsive behaviour when emotionally upset—were related to chronic anger expression in offenders across different countries (i.e., Italy and Australia) ([73]). Other studies focused on the link between self-esteem and violent acts showed how offenders reported lower levels of self-esteem than community participants, as well as greater levels of emotional nonacceptance and hostility. In fact, emotion dysregulation is considered as a mediator between low self-esteem and physical aggression, anger and hostility ([24]).

Linked with emotion regulation and self-esteem is the stigmatisation process, which relates to both victims’ and offenders’ interactions with the rest of the community. The stigmatisation process occurs when “human differences are distinguished and labelled; these labelled persons are subsequently linked to undesirable characteristics (negative stereotypes); those labelled are then separated into ‘us’ and ‘them’ categories; and finally, labelled persons experience status loss and discrimination that leads to unequal outcomes” ([42]). For stigmatisation to be fully realised, “power must be exercised” ([42]) to allow these processes to unfold. Rooted in normative judgments and carried out through interactions, the stigmatisation process reduces the person from a complex whole to a single tainted trait. Due to its pervasiveness, this process may have repercussions in many aspects of a person’s life: the stigmatised person tends to hold the same beliefs about their identity, leading to convince themself that they actually fit the label attributed to them ([25]). From a social construction approach, different scholars focused on how stereotypes and, more in general, the social construction of the roles of victim and offender, interact and affect the phenomenon of GBV. According to [53] ([53]), social constructionism focuses on examining the ways in which people build their theories and representations about reality, which then become objective for them. These implicit theories shape how people define the world, interpret events, make choices and interact with others—particularly in contexts like GBV, where power dynamics and social norms play a significant role.

In this regard, Nicolson shows how researching and working with GBV implies considering gender–power relations, as well as moral, legal and social perspectives on the phenomenon ([49]). The author stresses the necessity of addressing and paying attention to all those ‘implicit theories’ coming from the discourses made by the community about GBV. Following this perspective, she addresses the uses that the different roles of the community make of these narrations, showing how they can be configured as ‘coping strategies’ as well as defences against ‘thinking’, which might reduce the impact of advocacy ([50]). In continuity with her research, many other studies focused on how stereotypes and discourses can impact the phenomenon of GBV and IPV. Bates et al., for example, show how the persistence of these gendered stereotypes within the narrative on GBV can have negative effects both on societal responses to violence and on policy and practice development ([7]). Furthermore, other negative effects can be detected in relation to the victimisation process, a phenomenon likely to lead to issues on psychological ([52]), socio-economical and sanitary levels ([3]). As shown by the studies mentioned above, the discourses performed by people regarding GBV and IPV play a pivotal role in defining these phenomena. Those discourses have different repercussions on the experiences people have about this topic, as well as on the interventions and policies made to contrast it. Despite this rich literature on risk factors, prevalence and consequences, research using narrative approaches remains scarce ([57]; [16]; [60]). Indeed, few studies have investigated how victims, offenders and experts construct and portray their roles and experiences through narratives, where the direct perspectives of the individuals involved in these situations are often excluded from knowledge construction, but represent a highly valuable aspect—as also pointed out in ([11]; [12]). Our exploratory study contributes to this research strand by examining what happens in IPV situations, how they are described by those directly involved and how they impact the stigmatisation process for both victims and offenders, focusing in particular on emotion regulation and participants’ discursive modalities. By addressing such topics, we believe that this exploratory study could complement existing quantitative and epidemiological research, providing a qualitative understanding of the subjective and social constructions underlying GBV and IPV.

### 1.3. Research Objectives and Hypothesis

Given these elements, this exploratory research has two objectives: (i) to observe which discursive modalities are used by the groups of victims, offenders and experts to configure the roles of victim and offender in episodes of GBV and IPV, with an eye on terms related to emotion regulation; (ii) to offer operational reflections to widen the range of management possibilities of such phenomena, starting from anticipating the implication of using specific discursive modalities in terms of stigmatisation and health ([33], [34], [35]; [48]; [66]). In fact, our research aims to further deepen how the two main roles involved in IPV situations—victim and offender—are described. This is achieved by investigating the narratives of these roles directly provided by these same groups of people, as well as by those involved in the management of the phenomena, and by focusing on the emergence and use of terms related to emotion regulation. As deepened in the following section, these terms can be conveyed through different discursive modalities, representative of how natural language is used. The adoption of different discursive modalities can lead to different repercussions, like promoting a higher health degree or, conversely, fostering stigmatisation. Our research hypothesis is that all three groups adopt modalities that generate fixed and unchangeable narrations, given that the stigmatisation process confines the roles within fixed and immutable categories and given that it is the way in which language is used by the roles involved which allows for this process to be generated and observed.

Our research is based in Italy, Veneto Region, and involves a limited number of participants coming from a single institution working in countering gender-based violence. Considering this, we hope that this exploratory study will be a starting point for future investigations with larger samples of participants, which would be useful to increase the ability of institutions and professionals to continuously improve their work with victims and offenders.

## 2. Materials and Methods

### 2.1. Theoretical Background

From a theoretical perspective, our research is grounded within the interactionist and narrativistic paradigm ([5]; [67]). This is rooted in the work of [74] ([74]); [9] ([9]); [28] ([28]); and [44] ([44]), who state that reality is constructed by people through their interactions. The conceptual core of this perspective argues that the process through which people construct their identity depends on the way they interact. According to the narrativistic paradigm’s assumptions, we propose a discursive definition of gender-based violence, which includes different forms, such as IPV, domestic violence, etc.: “the set of discursive modalities that, using rhetorics linked to gender differences, characterise interactions and conduct that may lead both to the violation of norms/laws and to pursuing strictly personal goals whereby ‘the other gender’ (person/partner) is seen as an obstacle to be opposed/removed”. Since this definition focuses on how violence is discursively constructed in interaction, it is useful to observe the discursive processes within people’s narratives. For this reason, our methodological approach is based on textual data analysis, which allows for an observation of these discursive productions. One of the most used tools in this field of research is IRaMuTeQ, a digital interface based on R for statistical and multidimensional analysis of texts. We used IRaMuTeQ to obtain content data from the text collected, running Correspondence Factor Analysis (CFA), Descending Hierarchical Classification (DHC) and Similarities Analysis (SA). In particular, we focused on observing text occurrences that could be related to emotion regulation aspects, both when they were returned by the software as statistically significant and as non-significant.

Along with IRaMuTeQ, we applied MADIT (Methodology for the Analysis of Computerised Text Data), which is grounded in the analysis of natural language, understood as the medium used by humans to generate sense about the reality in which they live and interact ([6]; [68]). In fact, humans are able to create and narrate countless realities of sense about what they experience using natural language ([5]).

Through MADIT, these realities of sense ([23]) can be observed as sets of natural language use modalities called Discursive Repertories (DRs) ([6]; [68]). MADIT’s analytical procedure consists of 6 steps, as described in ([6]; [68]), and provides as output the DRs used (among 24 in total, grouped in three macro-typologies) and the resulting Dialogic Weight (dW). The typologies represent the direction of the narrative:Generative DRs open up scenarios whereby the narrative about gender-based violence can be joined by elements other than just the phenomenon or the specific violence situation such that there are multiple storytelling possibilities;Stabilisation DRs close off that possibility, making all of a person’s discourses and actions revolve around the gender-based violence phenomenon/situation in only one narrative;Hybrid DRs foster the generative or stabilisation direction based on the DRs they link with.

The contribution of each DR used in a person’s discourse indicates whether the narrative moves in a more generative or more fixed direction and therefore reflects the degree to which change is possible. This contribution is quantified through the Dialogic Weight (dW), which serves as a measure of the narrative’s configuration, ranging from 0.1 to 0.9. In other words, answers with low dW (tending to 0.1) mainly consist of Stabilisation DRs and trigger the start or retention of a typisation process. Conversely, answers with high dW (tending to 0.9) are mainly characterised by Generative DRs and hinder the rhetoric of typisation and stigmatisation. In order to obtain DR frequencies and dW values, we used D.I.Ana (Dialogical Interface for ANAlysis) 2.0 text analysis software ([67]).

As stated above, different discursive modalities (or DRs) can have different pragmatic repercussions, with some that foster health and others that can lead to stigmatisation. Regarding the latter, the discursive modalities involved in the stigmatisation process tend to anchor the roles to a judgmental label, reducing the variability of different possible narratives. This limitation applies both retrospectively and anticipatorily so that past choices and behaviours, as well as those imagined or assumed to occur in the future, are interpreted through the lens of this assigned label ([70]). Both roles contribute to constructing the phenomenon: one assigns a label to the other while perpetuating the label assigned to them. Their behaviours then align with these imposed identities that are developed by following stereotypes. This process also concerns other community roles, from citizens to professionals directly involved in contrasting the phenomenon (psychologists, social workers, law enforcement, lawyers and politicians).

Building on this theoretical foundation, our study adopts an integrated approach combining MADIT with IRaMuTeQ to enable a more detailed examination of GBV and IPV narratives. While some previous studies have applied either qualitative analysis methods or text analysis software separately—e.g., the works of [60] ([60]) and [16] ([16])—our research leverages both tools together, conceptually aligning with the interpretative possibilities discussed by [55] ([55]) in linking IRaMuTeQ and Discursive Textual Analysis for qualitative research. In our pilot study, this integration provides a more comprehensive view of discursive patterns and captures nuances of victim and offender narratives. In particular, the text’s observation made it possible to describe the actual configuration on IPV and the outputs stemming from the research process to propose potential management possibilities to change that configuration.

### 2.2. Data Collection and Participants

For data collection, we designed three different open-ended questionnaires, including one for each group of participants: victims of gender-based violence, offenders and experts. Questions followed two specific aims, as shown in Table 1. Following these aims, the 3 questions administered to the group of victims asked them to describe themselves, an offender, and how an offender would describe them as victims of gender-based violence. Similarly, the 3 questions administered to the group of offenders asked them to describe themselves, a gender-based violence victim, and how the latter would describe them as offenders. Differently, the 2 questions administered to experts asked them to describe both the roles of victim and offender in gender-based violence episodes (see Appendix A).

As already mentioned, participants in our research included the actors taking part in GBV and IPV episodes: victims, offenders and experts. Participants were chosen from the group of people who helped in this study and professionals from Fondazione “Eugenio Ferrioli e Luciana Bo Onlus”[note 1], which supported this research. The foundation was also adopting (and still adopts) the same theoretical and methodological assumptions of our research (Dialogic Science) in their daily interventions with both victims and offenders. We involved all the roles available in the foundation at the time the research was conducted: the total sample included 21 victims, 6 offenders and 8 experts (4 women and 4 men). The group of experts comprised all types of professionals involved in managing gender-based violence episodes and taking care of those involved, such as psychologists, lawyers, social carers, etc., employed in the foundation. The sample size is tied to the discursive focus of the study. During data collection, all participant responses were systematically analysed and coded for the DRs, meaning that each recurring pattern of language or role description in the narratives was identified and labelled. Saturation was considered achieved when adding further participants did not produce new narrative patterns or increase the variety of DRs observed, indicating that the sample captured the full range of discursive configurations relevant to victims, offenders and experts. As an example, in describing the role of victim, the victims’ answers began to show the same DRs as every other victim in relation to very similar contents and themes every 3–4 answers. We collected 89 answers, for a total of 1100 occurrences (with 262 hapaxes) and 119 DRs.

## 3. Results

Starting with MADIT’s results, Figure 1 shows the distribution of DR typologies used by each group of participants to describe the roles of victim and offender, respectively.

Table 2 reports the most used DRs and the value of dW for each group, again divided for the roles of victim and offender.

Regarding the role of victim in gender-based violence, the group of victims is the only one using Hybrid DRs to describe that role; it is also the group that employs the lowest amount of Generative DRs. The group of experts shows a wider narrative range, using more Generative DRs and fewer Stabilisation DRs, which indicates a more flexible way of describing the victim role. The group of offenders is the one that restricts the configuration of the victim’s role the most (through the Certify Reality DR). The quantitative results presented in Table 2 reinforce such qualitative interpretation. For instance, the marked use of the Certify Reality DR when offenders describe the victim’s role (50%) confirms that their narratives tend to restrict and narrow the victim’s representation. Similarly, when describing the role of offender in GBV episodes, victims represent the group that bounds most the configuration of this role, using the lowest amount of Generative DRs and the highest rate of Certify Reality. Indeed, this mimics what has been previously pointed out for the group of offenders: the very limited use of Generative DRs by victims when talking about offenders (e.g., only 4.55% Description) mirrors their more constrained portrayal of the offender’s role. The χ^2^ results from the CFA (Table 3 and Table 4) further support these interpretations: forms such as ‘to feel’ and ‘to be’ are positively correlated with victims’ self-description, demonstrating that these forms are crucial in their narratives. By contrast, experts show a wider narrative range, combining more Generative DRs with Hybrid ones. This pattern, further reflected in the higher dW values (0.66 for the offender’s role), suggests a more flexible and multidimensional understanding of both roles.

Regarding the statistical and multidimensional analysis conducted using IRaMuTeQ, Table 3 and Table 4 show the results of CFA for the roles of victim and offender, respectively. Each table is divided by group and reports relevant occurrences (with the untranslated Italian term in brackets) and the related *chi squared* value. We also included some forms that are not statistically significant, either because they appear significant for other groups or could have been considered ‘typical’ of those groups, but are actually scarcely part of our participants’ narratives. By integrating the χ^2^ results with qualitative interpretation, we can directly observe which narrative forms are statistically salient for each group, providing quantitative evidence for the patterns discussed above.

In relation to the victim’s role, the group consisting of women who are victims of gender-based violence self-assigns this role to themselves, emphasising ‘to be’, especially feeling like it (form ‘to feel’). Other forms such as ‘helpless’ and ‘to react’ have negative correlations, meaning that they are scarcely part of their narrative as women who are victims of IPV. Offenders do not use particularly salient words in relation to the role of victim other than the form ‘to have’, which, however, occurs in association with non-significant terms like ‘helpless’ and ‘scare’. Experts instead provide a content pattern opposite to the victim’s group: they adopt forms linked to fear and helplessness in reacting but do not restrict women only in the role of victims as the negative correlation for the terms ‘to be’ or ‘to feel’ represent.

Concerning the offender’s role, the group of victims pictures the offender as being—inherently—violent, as shown by the positive correlation with the terms ‘to be’ and ‘violent’ itself. Offenders also describe themselves as violent, but to a lesser extent. In addition, the form ‘violent’ is joined by the verb ‘to do’ and not ‘to be’—thus, it is not something inherent but acted. Both groups show a negative correlation with the form ‘to be able’ and a positive correlation with the opposite one, ‘unable’: this accounts for the role of the offender to be depicted as someone that is not able or in the condition to do something, for example, control themself or behave differently. Conversely, the group of experts do not describe offenders as inherently ‘violent’. Indeed, the word ‘violent’ has a negative correlation in their answers: this means that experts do not necessarily use such attribute to describe IPV offenders and shows a more neutral or multidimensional perspective.

Figure 2 and Table 5 show the results of the DHC. Figure 2 indicates how much each cluster accounts for the total dataset, while Table 5 shows the text forms associated with each cluster. As for Table 3 and Table 4, in Table 5, we reported some statistically non-significant forms which could have been considered ‘typical’ of the two roles but are not frequently used by the participants in our sample.

DHC provides two clusters, covering 58.30% and 41.70% of the total dataset, respectively. The two clusters pertain to and distinguish the two roles under investigation, with different text forms for each one. Such proportions support the insight that offender-related discourse is slightly more dominant in the sample, reinforcing the interpretation of role salience across participants. The role of offender is associated with ‘person’, ‘weak’ and ‘violent’, confirming the observation that participants associate offenders primarily with these judgmental and personal descriptors. Differently, words related to emotion regulation like ‘fear’, ‘anger’ and ‘emotion’ are present but not statistically significant. Therefore, that role is indeed connoted with judgements, but only marginally through emotional characteristics. Instead, the victim’s role is associated—besides the term ‘victim’ itself—with terms most closely linked to emotion regulation, such as ‘fear’, ‘guilt’ and ‘helpless’ (with the latter to a lesser extent). This supports the interpretation that victims’ narratives emphasise emotional experience and responsiveness. As it can be noted, the form ‘situation’ appears as not significant (with the lowest *ρ*-values): indeed, looking at participants’ answers, the ‘violent situation’ is an element scarcely referred to in narratives related to victims.

Lastly, Figure 3 reports the results from the SA.

The SA generated five macro-areas: two predominantly related to the role of victim, and two to the role of offender. Regarding the first role, both macro-areas are reported as relevant forms associated with emotion regulation (‘helpless’ and ‘unable’; ‘guilt’ and ‘fear’). In relation to the second role, however, only one of the two macro-areas shows terms linked to the construct (‘anger’ and ‘aggressive’). By linking the structure of the SA macro-areas with content analysis, it is evident that the first two macro-areas (victim-related) correspond to the strongest emotional content in participants’ discourse, confirming that victims’ narratives emphasise emotional experiences. The offender-related macro-areas, in contrast, show fewer emotion-linked forms and more judgmental terms, quantitatively supporting the qualitative observation that offenders are described in more restricted or action-focused ways.

These quantitative patterns can be further understood by considering the broader social and institutional contexts in which participants’ narratives are embedded. Indeed, this pilot study was conducted in a specific Italian region, and participants’ narratives could be influenced by regional norms, local attitudes toward gender roles and the protocols of anti-violence centres. Following this, we will mention such aspects where relevant. Overall, the role of victim in gender-based violence is described as being strongly closed to different possibilities other than the one depicted. In fact, MADIT’s results show a widespread use of Stabilisation DRs by all three groups of participants, leading to a low overall dW (0.18 dW). The most used DRs are Certify Reality and Judgement, both pertaining to the Stabilisation typology. The former poses the description as a matter of fact, absolute and unchangeable; the latter shares the same direction while combining it with the use of strictly personal criteria and values, thus morally connoting the description.

Deepening the description provided by each group, participants in the victim group tend to convey the “guilt self-attribution” ([10]; [47]), i.e., they appear to limit their role around the chance to be guilty of the violence against them, using expressions like


*“The person loses his dignity, feels guilty. When she has problems with her husband, she avoids them”.*


This is supported by IRaMuTeQ’s output, where statistically significant forms outline an association between ‘being’ and ‘feeling’ like a victim and guilt (see Table 3 and Figure 3).

In the above example, the emotional feelings of loss of dignity and sense of guilt tend to function as a stigmatising pivot within the narrative: the victim could start describing themself predominantly as the one deserving violence, and the offender may draw on this attribution in ways that could contribute to the continuation of violent behaviours. Referring to the social construction of the roles, as mentioned in ([8]), the sense of guilt is linked to the roles the people have in society and, as in the example, individuals who feel guilt might state that they “should have known better” instead of making a specific decision.

These patterns may also reflect regional cultural norms around gender roles and expectations, which can reinforce the internalisation of guilt in victims and the perception of passivity in the IPV context.

In addition, such a narrative may also influence the victim’s willingness to seek help. Specifically, the second part of the response suggests that the victim might become less inclined to take action or assume responsibility for their situation, potentially leading to avoidance and a reduced likelihood of reaching out to support services. Therefore, the combined use of Stabilisation DRs and such contents may contribute to tendencies in which victims keep talking about themselves according to these narrations, possibly entering into a stigmatisation process as described in ([17]). Looking at the offenders’ group, they depict the role of victim from a strictly personal position, as suggested by the percentage of Stabilisation DRs used. Thus, their descriptions appear grounded on thoughts and beliefs that may not be shared or validated by other groups, adopting expressions such as


*“[the victim is] weak, insecure”.*


Using such a combination of DRs and terms, it is possible to anticipate that an offender could frame their position in ways that could be perceived as legitimising behaviours associated with GBV and IPV ([63]), maintaining a relationship of “power abuse on the woman” ([2]; [21]).

Similar terms as the ones reported in the previous example are also employed by the group of experts, although with a different significance. An example is the following answer:
*“[the victim is] helpless, succubus, a frightened person”,*
in which the text conveys support and comfort to the victim. However, since these terms are mostly used with Stabilisation DRs, the description about the victim appears relatively closed and limited, which may suggest that interventions could focus primarily on protection (see also ([46])). As much as victim protection is a core concern in these situations, only focusing the interventions on this aspect could limit opportunities for the victim to develop skills that can enable them to identify critical interactions and behaviours acted out by their partner, potentially helping them to anticipate and respond to risk. Conversely, in promoting such skills, it might also increase the likelihood that victims will take more ownership in promptly contacting dedicated support services. It is also important to consider that experts operate within the institutional guidelines of anti-violence centres that may implicitly encourage a tendency towards Stabilisation DRs, emphasising protection over empowerment. The specific policies, procedures and cultural attitudes within centres could thus shape the DRs used by experts.

Such descriptions, which tend to anchor the victim only in that role, could foster stigmatisation and risk to reduce the likelihood of change ([37]); however, effects should be interpreted as tendencies rather than definitive outcomes. Unlike the other two groups, this direction is partly counteracted by the experts, who adopt Generative DRs for ¼ of the total. Moreover, unlike the group of victims, they do not associate the role of victim of GBV as something a person inherently is or feels (see Table 3 and Table 4). As it can be observed in those texts, the cluster ‘situation’ is not relevant. This is due to the fact that victims are not described as in the specific situational context of IPV, but only in terms of personal features such as fear, guilt or helplessness, generalising those features across situations beyond IPV. This tendency may be influenced by the context of IPV management in Italy, where victims’ experiences are often framed in generalised emotional terms (also in reporting procedures and protocols).

Following with the role of offender in GBV episode(s), its overall description appears to resemble the victim’s in terms of the DRs used. Indeed, MADIT’s output shows a predominant adoption of Stabilisation DRs, although it is generally lower in comparison to the role of victim. This results in a mean dW (between the three groups of participants) of 0.38. The most frequently used DR is still Certify Reality, followed (with similar rates) by Judgement and Description. The latter DR relates to the use of commonly intelligible elements, devoid of personal beliefs and theories, thus providing a shareable picture of the role. The presence of the Description DR is what makes the overall description of the role of offender less closed to other possibilities, albeit that direction remains prevalent.

Answers from the victim group are the most representative of this direction, inasmuch as they are conveyed through Stabilisation DRs for over 90% of the total. A prime example is the following text:


*“[the offender is] overbearing, impulsive, verbally aggressive”.*


In this answer, for example, the attribute ‘impulsive’ assumes the characteristic of a personality trait: on one hand, this justifies the offender as being unable to do otherwise and to have control over their behaviours; on the other hand, it removes the responsibility of their actions. This consideration is also supported by the diverging correlation of the forms ‘unable’ (positive) and ‘to be able’ (negative)—also present in the group of offenders: the offenders are pictured as people that are unable to do things in a certain, different way from what has been performed (which may be the violent act itself or, more in general, the ability to control themselves).

Content wise, victims’ expressions about the role of offender tend to link psychological and emotional characteristics to the offender’s behaviours, as if these would be inherent and prerogative only of that role (and not of others) ([61]). This way, they configure both characteristics and conducts as unchangeable facts, in turn risking reducing the possibility for them to assume (even as a result of a recovery path) other roles (such as the one of father, worker, etc.) ([37]; [61]).

The group of offenders describes their role in a slightly more generative way, although Stabilisation DRs cover almost ¾ of the total. This direction also appears in the content dimension, inasmuch as they refer to ‘violence’ only as something they do and not something they are (see Table 4).


*“I have always got everything in life, what I want I get, I don’t care how other people are/what they want, it’s me who has to be well”.*


This is the opposite from the picture given by the group of victims, who appear to judge them as inherently violent and unable to behave differently. Independently of this, however, the adoption of Stabilisation DRs reduces the chances of changing the trajectory of the narrative toward health. Taking the above answer as an example, using such modalities, the scenario of “it’s me who has to be well” cannot take place: certain that the offender will always get everything and that the partner is irrelevant, the offender is unable to anticipate the repercussions of their own actions, which have already led them to the charge and trial, and may even lead them to conviction and imprisonment. Indeed, from their narratives, it appears that offenders are often unaware of the legal consequences of their actions, an aspect that may reinforce a focus on personal justifications rather than reflection on potential changes.

Lastly, a different scenario is depicted by the group of experts: indeed, among the three groups, experts are the only ones adopting Hybrid DRs and the same rate of Stabilisation and Generative DRs. This combination of DRs allows them to provide a much more shareable and commonly agreeable description of the role of offender, thus opening the door to wider possibilities of interaction and, in turn, of change through intervention pathways. A representative example of this direction is the following sentence, in which the Hybrid DR of Evaluation is used:


*“[The offender] is angry, disappointed, scared by the fact that he cannot accept the end of the relationship or the loss of his job due to anger in the workplace”.*


Even though the terms used relate to psychological and emotional features of the offender (‘anger’ and ‘scare’)—similar to the ones used by the group of victims—in this answer, the criteria used to depict that role are made explicit and provide useful elements to be exploited by other interactants (colleagues, but also the users of an IPV service) to widen the description and steer it towards different scenarios. Furthermore, as for the role of victims, the group of experts do not seem to attribute intrinsic characteristics of violence to the offender’s role.

The elements used to describe them pertain to difficulty in managing emotions, emotional non-acceptance and poor distress tolerance, thus relating to low emotion regulation ([72]). This can be confirmed by the fact that clusters related to emotion regulation, such as ‘scare’, ‘anger’ and ‘emotions’, are not frequently used by the participants to describe offenders.

## 4. Discussion

### 4.1. Practical Implications

From the findings of our exploratory research, it appears that both roles under investigation—victim and offender—tend to be bound within narrow configurations. As a generalised tendency, both the victim and the offender roles share the risk of entering into a process of stigmatisation; however, they differ on the specific practical repercussions.

Victims tend to be depicted through terms related to their emotions and emotional experiences, conveyed mostly with Stabilisation DRs such as Certify Reality and Judgement. In this way—and in line with what has been suggested in ([8])—as victims, the fear and guilt experienced risk becoming unchangeable parts of their narrative as they are used by victims to justify the violent actions of their partners and not to change their situation. In turn, this reinforces them being and feeling only as victims. As a practical repercussion of such spiralling tendency, it is possible that victims downsize the violent behaviour of their partners, tell themselves that they are the ones that should “behave better” ([8]) and reduce the chances of reaching out for help to friends, family and services.

Offenders are mostly portrayed with moral and judgmental terms rather than with emotional ones. However, these are still conveyed through the Stabilisation DRs of Certify Reality and Judgement. While victims tend to describe offenders as inherently violent, and offenders state that they ‘act violently’, both groups depict the role as someone unable to control themself and their behaviours in a narrative where the possibilities to do things differently risk being reduced. In this way, as two possible practical implications, offenders may not only be implicitly legitimised to perpetuate their violent behaviours (since “offenders are overbearing, impulsive” and “that’s how they are, and they cannot act differently”), but they may also fail to consider and anticipate the repercussions of their actions for themselves, especially on the legal side, and for their victims in terms of psychological trauma, stigmatisation and everyday life restrictions.

Considering such configurations of the victim and offender roles, experts may also face practical challenges, in particular in their interventions. Indeed, they are likely to encounter and deal with people adopting narratives similar to the ones described above: victims may struggle to see alternatives to such identity and to the stillness related to the emotions experienced, while offenders may fail to recognise both the need and possibility for different behaviours. Thus, as a practical implication, experts may need to design and deploy more complex and demanding intervention strategies in order to promote change, inasmuch as they would have to consider creating opportunities—both for victims and offenders—to engage in other possible and different roles that they could inhabit in their lives ([61]).

### 4.2. Reflections for Intervention

We now discuss our insights and reflections for intervention; a first aspect to leverage is precisely the set of DRs used by the group of experts. Indeed, we observed how they do not configure the role of victim as something intrinsic to the person, nor the role of offender as violent ‘per se’—contrary to what victims and offenders themselves do. This is crucial for health-promoting interventions and particularly relevant when considering the Italian institutional context, where services for victims of IPV follow specific protocols and guidelines that influence how experts should or have to interact with users: indeed, from the differences between the groups’ answers, we can state that configuring roles’ attributes as emotional features/personality traits or as acted behaviours is a reflection of the discursive modalities used to describe them, which may differ. Adopting Generative DRs and carrying out interventions focused on making victims and offenders use more Generative DRs could, for example, foster a firmer responsibility towards IPV: victims would be more aware of the situation and have the tools to identify interactive indicators of IPV in their relationships with their partners; offenders could recognise and take accountability for their behaviours, using the events occurred to anticipate the repercussions of their choices and to act differently in the future.

Conversely, we can suggest that in using Stabilisation DRs—as they partly do—and terms solely related to a psychological–emotional dimension (such as ‘scare’, ‘helpless’, etc.), both victims and offenders may tend to become strictly bound within a stereotypical dimension characterised by moral attributes associated with the common labels of such roles. Furthermore, policy-driven constraints, such as mandatory reporting or intervention procedures, may inadvertently reinforce these stabilised narratives if not critically reflected upon by experts. Also, another critical repercussion that could arise if experts mostly adopt Stabilisation DRs is that their questions and suggestions to victims or offenders may negatively influence the overall interaction: in fact, they could create a ‘self-feeding’ process, where the user might align their narratives according to the expert’s, possibly legitimising and reiterating those narratives. This process could have implications not only in terms of potential stigmatisation, but also on the legal level. Indeed, keeping in mind that gender-based violence scenarios often lead to legal trials, one example is that the reliability of the testimony (both of victims and offenders) could be affected by the interaction occurring within the expert/service ([38]), inasmuch as such testimonies are a representation of a particular narrative (provided to certain discursive modalities). Therefore, considering that experts’ narratives are also a product of the discursive modalities employed, they should remain maximally adherent to the answers and accounts provided by victims and offenders, trying not to overwrite them with their own. This would be effectively achieved by recounting users’ narratives through the Description and Consideration DRs, sticking thoroughly to the narrative answers provided by users and methodically asking themselves if the questions they ask victims and offenders originate from those answers or from their personal theories about the victims and offenders. These practices should also be informed by broader cultural and institutional factors, such as community expectations regarding gender roles, regional norms about IPV and the legal responsibilities of service providers, which all shape how narratives are expressed and interpreted.

In light of this, the results of our exploratory study regarding the social construction of the roles involved in GBV and IPV could be used as an opportunity to further reflect on the praxis adopted by field experts and dedicated services. As already argued, specific DRs have peculiar pragmatic implications for these roles: particularly, in the context of clinical intervention, the adoption of Stabilisation DRs may tend to maintain the current narratives as ‘difficult to change’ and potentially reduce the manifestation of different narrative possibilities. In this way, the impact on the rehabilitation path could be limited: victims and offenders’ narratives about themselves might reflect pronounced tendencies to remain within such roles, and the responsibility of change would be fully delegated to experts—or, especially for offenders, be difficult to define ([39]). Since the DRs used so far and their impact on gender-based violence (GBV) and intimate partner violence (IPV) have been described, experts can now propose new narratives for both victims and offenders—narratives that promote personal responsibility. These new narratives should shift the focus away from viewing the roles involved (victim and offender) as being passively dependent on experts, moving toward a perspective where individuals can anticipate the repercussions of their actions and find a way to change rather than being confined to predefined roles. This new configuration also leaves room for other possibilities and proposals to emerge from the roles involved, allowing for different directions than those previously taken.

Conversely, the adoption of a higher rate of Hybrid and Generative DRs would make it possible for experts to promote change in the way victims and offenders narrate themselves and each other. In turn, such narratives could open up more possibilities for the development of new and more effective intervention strategies. Considering this, experts may focus on employing interactive modalities (DRs) that ‘push’ users to position themselves as community members actively contributing to a shared management of critical issues regarding this phenomenon. Such narratives could be aimed at promoting the explication of the criteria for which a given thing is asserted (e.g., the second example answer above) and/or at the use of evaluations and descriptive elements that make the narrative shareable. Both of these strategies could move the interaction within the clinical intervention towards previously unsought narratives, thus offering users the possibility to assume other roles than the one played so far (and potentially leaving it).

## 5. Conclusions

Our exploratory research aimed to provide insights into how the roles directly involved in gender-based violence and IPV episodes are narrated by the actors involved in these situations—firstly, the victims and offenders themselves, and secondly, experts who manage such situations. Albeit with a limited sample, involving these three groups of participants allowed us to obtain a broader picture of how gender-based violence is socially constructed ([34]), considering multiple perspectives on the phenomenon and exploiting which specific discursive modalities are employed by the main roles involved.

The analysis of the narratives provided by participants resulted in a widespread use of personal theories and references by all groups involved: both the roles of victim and offender are described through Stabilisation DRs in a way that tends to not consider other possibilities and scenarios different from the ones already adopted, falling into stereotypical labels. Such a process could become particularly critical if it takes place in the context of services working with victims and offenders since it could risk promoting processes of victimisation for the former and stigmatisation for the latter ([37]).

We also offered some insights regarding the management and rehabilitation paths for the two roles under investigation. We highlight the possibility and usefulness of adopting more Generative DRs and considering not only the psychological characteristics of the actors involved—as suggested and deepened by the literature—but also all aspects that characterise the structure of a stereotypical narrative ([1]; [76]). In particular, thanks to the application of MADIT, it was possible to recognise various types of narratives. Since the analysis starts with a focus on “how the narration unfolds” and later focuses on “what is said in the narration”, the interventions can be made to change the way the involved actors narrate themselves. For example, if the term ‘impulsive’ is used to define the offender, the interventions could focus on how this term is used. Is it used to justify the behaviour or to describe a past action in order to take responsibility? Furthermore, if the term ‘succubus’ is used to define the victim, is it used to justify the state of things in a specific case or to generalise all victims? The intervention aimed at the people involved consists of changing the way the involved actors narrate the roles, shifting to a more Generative narration. The use of Generative DRs becomes a useful tool to restrain the implications and possibilities of IPV events to be predominant in a person’s biography instead of one of many life paths (both for victims and offenders). Thus, the health degree changes as much as the generative value of the narratives—which is possible to measure, as it is and in its variation, with MADIT: the lower the dW, the lower the health degree measured; conversely, a high dW indicates a higher degree of health.

Moreover, since other roles besides experts are involved in managing the phenomenon—such as public services and institutions—it would be useful for experts to be aligned with the intervention strategies that must be deployed. In fact, it could be detrimental and inefficient if experts made efforts not to relegate the victims to their victim status, while the institutional roles involved did not act in the same manner, as this may lead victims to revert to the initial stereotypical narrative. An example of this is given by Russell, who highlighted how police officers’ perceptions of potential danger (likelihood of past or future harm to their partner) and victim credibility are influenced by disputant gender and sexual orientation ([59]). To achieve this goal of alignment, it could be beneficial for experts to provide the other actors involved with precise and applicable intervention guidelines. As a result, public services and institutions could build different intervention strategies based on an interdisciplinary perspective where all community members are involved in promoting health in victims and offenders ([69]; [70]), opening up, for example, a restorative perspective on the management of this phenomenon ([39]).

Our pilot study does not come without limits. The first is related to the sample size and composition: it is limited to only 35 unevenly distributed participants, with a marked prevalence of victims (approximately three times higher) over offenders and experts. Linked to this, another limitation lies in the study’s focus on a single Italian institution. As briefly mentioned above (Section 2.2), considering the relevance of the topic, we decided to exploit the availability of the foundation and the contribution provided by the participants, even if their number was small; in particular, at the time, it was difficult for us to reach other subjects pertaining to the offenders group, and few experts were available in the research context. However, we acknowledge that these research limitations do not allow for the generalisability of our results and could bring specific contextual biases linked to, for example, the institution and its location. Social variables were not explored during the interviews, which raises the possibility of unaddressed confounding factors. However, it is worth noting that participants did not spontaneously refer to such variables in their narratives, suggesting that these dimensions may not have been relevant to the discursive construction of roles in this context. To address these issues, future research needs to enlarge the sample size, both overall and relative to the groups of offenders and experts, in order to increase the significance of the findings and observations. Moreover, it would also be useful to involve multiple institutions from other Italian regions in order to reduce possible contextual biases. Once this is carried out, it would be valuable to expand the research to international contexts, analysing differences and similarities between them. A parallel line of research would be to conduct a longitudinal study, assessing the long-term impact of the discursive modalities adopted by victims and offenders on both recovery pathway and stigma perception. This could lead to more in-depth insights regarding the relationship and possible correlations between IPV narratives and psychological outcomes. Lastly, in our study we did not collect narratives from other community members who still interact and play a role in the way IPV is shaped, such as the families and friends of both victims and offenders ([41]). In fact, by interacting with victims or offenders, they could reinforce the stigma or, conversely, act to counteract the phenomenon and promote health. Therefore, another research line that could be explored in order to overcome such limitations and understand the phenomenon in its complexity would be to further widen the participant sample by reaching out to the circle of acquaintances of the individuals involved.

## Figures and Tables

**Figure 1 behavsci-15-01229-f001:**
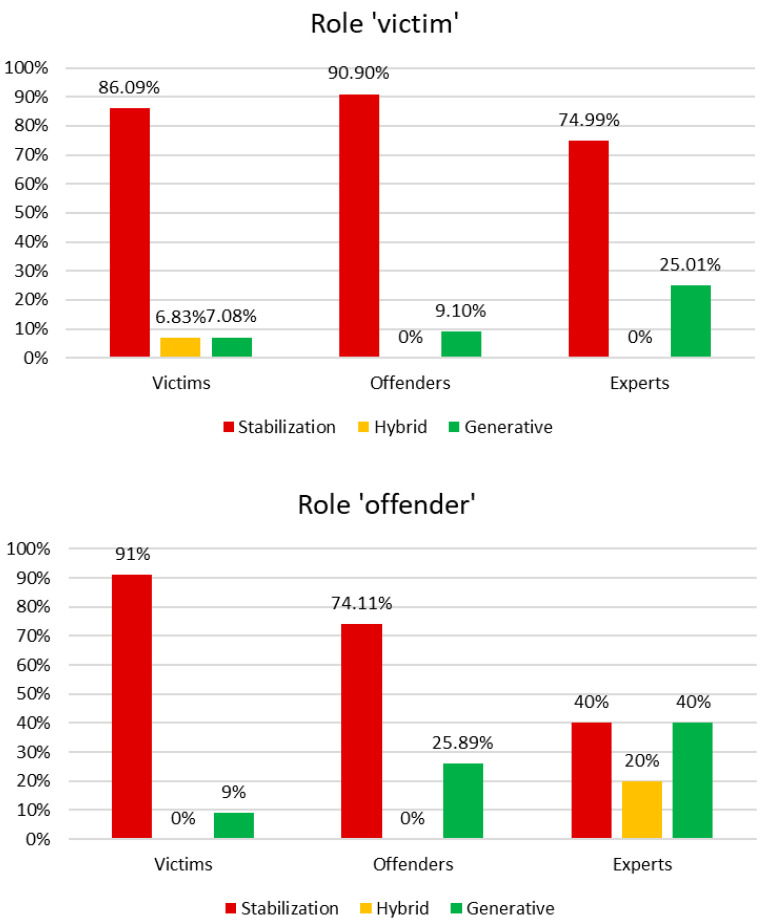
Distribution of DR typologies among participants’ answers for the roles of victim and offender, respectively.

**Figure 2 behavsci-15-01229-f002:**
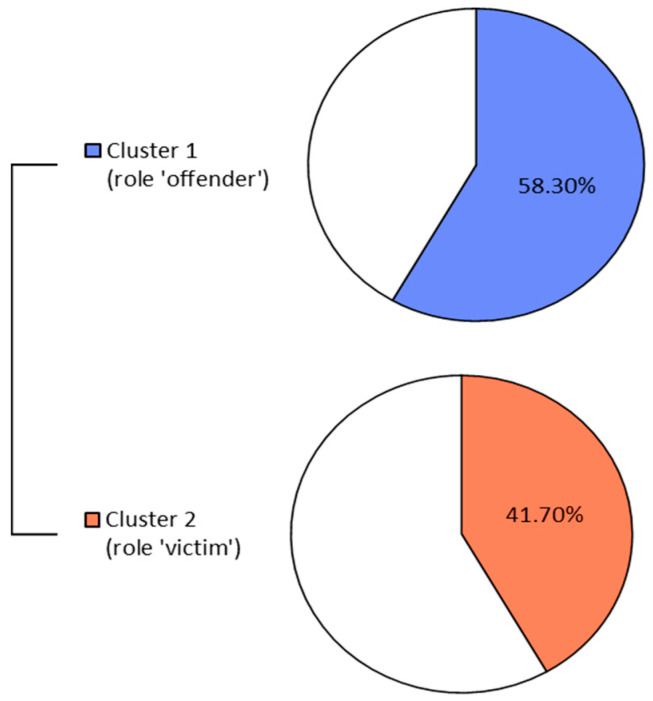
DHC results: cluster subdivision.

**Figure 3 behavsci-15-01229-f003:**
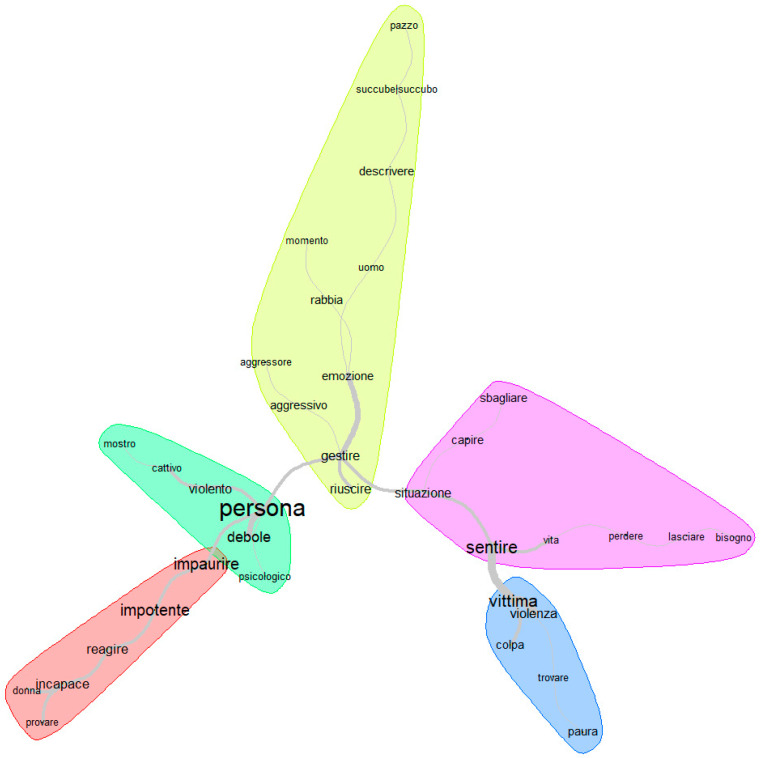
SA results (in the Italian language, as the figure was directly extracted from IRaMuTeQ).

**Table 1 behavsci-15-01229-t001:** Investigation areas and their related aims.

Investigation Area	Specific Aim
Role of ‘victim’	To outline how a person who has undergone violence is narrated and talks about themself.
Role of ‘offender’	To outline how a person who perpetrated violence is narrated and talks about themself.

**Table 2 behavsci-15-01229-t002:** Most frequent DRs for each group and related dW for the roles of victim and offender, respectively.

	Role of ‘Victim’		Role of ‘Offender’	
	DRs	dW	DRs	dW
Victims	- Certify Reality (30.13%) - Judgement (25.38%) - Generalisation (14.35%)	0.14	- Certify Reality (40.91%) - Judgement (31.82%) - Description (4.55%)	0.13
Offenders	- Certify Reality (50%) - Judgement (37.50%)	0.1	- Certify Reality (37.50%) - Judgement (34.82%) - Description (25.89%)	0.35
Experts	- Certify Reality (33.33%) - Description (25%) - Judgement (16.67%)	0.3	- Description (40%) - Evaluation (10%) - Generalisation (10%)	0.66

**Table 3 behavsci-15-01229-t003:** CFA results related to the role of victim, divided by group of participants.

Victims		Offenders		Experts	
Forms	χ^2^	Forms	χ^2^	Forms	χ^2^
to feel (sentire)	1.6636	to have (avere)	2.8864	scare (impaurire)	1.5334
to be (essere)	1.082			helpless (impotente)	0.8882
				to react (reagire)	0.7162
to react (reagire)	−0.5127	helpless (impotente)	−0.2289	victim (vittima)	−0.5084
helpless (impotente)	−0.6178	scare (impaurire)	−0.2289	to be (essere)	−0.8447
scare (impaurire)	−1.1256	to be (essere)	−0.3852	to have (avere)	−0.967
				to feel (sentire)	−1.2972

**Table 4 behavsci-15-01229-t004:** CFA results related to the role of offender, divided by group of participants.

Victims		Offenders		Experts	
Forms	χ^2^	Forms	χ^2^	Forms	χ^2^
to be (essere)	1.2227	to do (fare)	0.7909	to be able (riuscire)	1.2749
violent (violento)	0.7909	unable (incapace)	0.2959		
unable (incapace)	0.2959	violent (violento)	0.2959		
to have (avere)	−0.4284	to be able (riuscire)	−0.3063	unable (incapace)	−0.4369
to do (fare)	−0.8835	to be (essere)	−1.0844	violent (violento)	−1.1136
to be able (riuscire)	−0.8835				

**Table 5 behavsci-15-01229-t005:** DHC results: text form for each cluster.

Cluster 1—Role of ‘Offender’		Cluster 2—Role of ‘Victim’	
Form	*ρ*	Form	*ρ*
person (persona)	0.00015	victim (vittima)	0.00011
weak (debole)	0.01735	to feel (sentire)	0.00182
violent (violento)	0.04839	(to react) reagire	0.00225
scare (impaurire) *	0.07226	(violence) violenza	0.00225
anger (rabbia) *	0.08018	fear (paura)	0.01430
emotion (emozione) *	0.08018	guilt (colpa)	0.01430
		to understand (capire)	0.01430
		to find (trovare)	0.03549
		helpless (impotente)	0.03995
		situation (situazione) *	0.06937

Forms marked with * are not statistically significant.

## Data Availability

The raw data supporting the conclusions of this article will be made available by the authors upon request.

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
