# Peer review of "Stigma and Emotion Regulation in Intimate Partner Violence: A Pilot Exploratory Study with Victims, Offenders and Experts"

_behavsci, 2025, doi:10.3390/bs15091229_

Round 1

Reviewer 1 Report (Previous Reviewer 2)

Comments and Suggestions for Authors

The authors have provided brief responses to the reviewers’ comments and suggestions. However, a significant methodological concern remains unresolved. Specifically, the textual or content analysis presented is overly complex and, at times, difficult to follow. Given the highly anticipated findings, namely, the victims’ experiences of fear and self-guilt, and the offenders’ expressions of anger and pathology, depicted by victims, offenders, and experts, I recommend simplifying the analysis to enhance clarity and accessibility. Furthermore, much of the material currently included in the discussion section would typically be presented as part of the results in qualitative research. I encourage the authors to revise this section accordingly to align with conventional qualitative reporting standards.

Author Response

We thank the reviewer for the constructive feedback. We tried to address all the concerns, hoping that the changes made will improve clarity and adherence to conventional qualitative reporting standards. Here follows a point-by-point reply to the comments.

However, a significant methodological concern remains unresolved. Specifically, the textual or content analysis presented is overly complex and, at times, difficult to follow. Given the highly anticipated findings, namely, the victims’ experiences of fear and self-guilt, and the offenders’ expressions of anger and pathology, depicted by victims, offenders, and experts, I recommend simplifying the analysis to enhance clarity and accessibility.

The Results section has been revised to simplify and clarify the presentation of the textual and content analyses, in order to make it easier to follow. Sentences that were previously overly complex have been restructured for clarity, and key findings regarding victims’ experiences of fear and self-guilt, as well as offenders’ expressions of anger and pathology, are now presented in what we believe is a more direct and accessible manner. 

Furthermore, much of the material currently included in the discussion section would typically be presented as part of the results in qualitative research. I encourage the authors to revise this section accordingly to align with conventional qualitative reporting standards.

Following the suggestion, a portion of the original Discussion section that described detailed narrative patterns and discursive repertories has been moved into the Results section. This includes descriptions of how victims, offenders, and experts construct the roles and how DRs and statistical outcomes support these interpretations.

Reviewer 2 Report (Previous Reviewer 1)

Comments and Suggestions for Authors

Thank you very much for following the advice, it has improved your initial work.

Author Response

We sincerely thank the reviewer for the feedback and all the valuable suggestion provided throughout the peer-review process, that allowed us to improve the quality of our paper.

This manuscript is a resubmission of an earlier submission. The following is a list of the peer review reports and author responses from that submission.

Round 1

Reviewer 1 Report

Comments and Suggestions for Authors

It seems to me a work done with great effort and dedication, but I think it is not for this journal.

The first and fundamental thing is the objective that pretends to cover a lot, victims (21), abusers (6) and professionals (8) are very small samples. Perhaps it should have focused on a group of participants and had a larger sample and centered the study on it.

In the theoretical framework general aspects of violence against women are raised, with a lack of literature that contrasts different statements too sharp and that other authors raise from another perspective or reasons (women report more days mentally affected than men each month) or the abuse of psychoactive substances or antisocial personality are more common in men, (Moyano and Ramos, 2007) is a very recent quote and other examples. It would have been fundamental if the objective is to delve into gender roles, stigmatization and health, the theoretical framework should be more focused on that and not so general.

It would have been interesting to give more packaging and potential to the methodology to use a mixed research (qualitative and quantitative). The open-ended questionnaires have not been peer-reviewed and their validity is weak.

The specific objectives are unachievable, too complex. Sometimes it is much better if they are simpler and can be researched and contributed to.

The results cannot be generalized and, being such a small sample, it is even dangerous to assert certain statements. 

The conclusions do not contribute anything new; we are aware that the legitimization of the abusers and the asymmetry of power is what gives them the power to abuse a woman who is their partner. Perhaps variables such as attribution of responsibility, minimization, etc. should have been evaluated in abusers and the results should have been seen and maybe how to work on it and the same with the victim. Regarding professionals, I think that the study suggests that those who work with people in the social field should constantly look in the mirror, see the prejudices, etc. that do not help us to be professionally objective, but the article does not provide solutions. 

The bibliography does not present major errors but there are some small flaws to improve, for example the journal in italics, etc.

Author Response

First of all, we thank the reviewer for all the valuable suggestions and inputs provided. Here follow a point-by-point response to the comments:

1. The first and fundamental thing is the objective that pretends to cover a lot, victims (21), abusers (6) and professionals (8) are very small samples. Perhaps it should have focused on a group of participants and had a larger sample and centered the study on it.

We understand the issue and certainly agree that our sample was limited. At the time the research was conducted, that was the whole number of participants available: we still thought to proceed in order to value their contribution and availability, considering the relevance of the topic. We now specified these elements within the limitations of the study, as well outline future research lines focused on each group. In line with this, we also changed the objective’s formulation.

2. In the theoretical framework general aspects of violence against women are raised, with a lack of literature that contrasts different statements too sharp and that other authors raise from another perspective or reasons (women report more days mentally affected than men each month) or the abuse of psychoactive substances or antisocial personality are more common in men, (Moyano and Ramos, 2007) is a very recent quote and other examples. It would have been fundamental if the objective is to delve into gender roles, stigmatization and health, the theoretical framework should be more focused on that and not so general.

We added the suggested references, as well as other ones, to enhance the literature background on the topic. We also changed the overall framework of the Introduction section to highlight more the specific topics rather than the general ones used before.

3. It would have been interesting to give more packaging and potential to the methodology to use a mixed research (qualitative and quantitative). The open-ended questionnaires have not been peer-reviewed and their validity is weak.

We thank the reviewer for this suggestion. In line with that, we added to the future development of our study the comparison with other methods. We also specified in the text that one of the criteria used for choosing our methodology is that it was (and still is) recognised and adopted by the partner Foundation’s experts for their consultations with both victims and offenders. So, as far as the questionnaires have not been validated, they have been approved by the Foundation’s experts.

4. The specific objectives are unachievable, too complex. Sometimes it is much better if they are simpler and can be researched and contributed to.

We simplified the formulation of both the general objective and the specific ones, in order to make them more understandable and allowing other researchers to align with them and deepen the exploration of those aims and questions with their research.

5. The results cannot be generalized and, being such a small sample, it is even dangerous to assert certain statements. 

We thank the reviewer for pointing this out. We certainly agree, and explicitly stated that the issues about the generalizability of our results are a limitation of the study that we need to address with future research. We also “softened” and downsized some of the statements provided in the Discussion, using more hypothetical forms.

6. The conclusions do not contribute anything new; we are aware that the legitimization of the abusers and the asymmetry of power is what gives them the power to abuse a woman who is their partner. Perhaps variables such as attribution of responsibility, minimization, etc. should have been evaluated in abusers and the results should have been seen and maybe how to work on it and the same with the victim. Regarding professionals, I think that the study suggests that those who work with people in the social field should constantly look in the mirror, see the prejudices, etc. that do not help us to be professionally objective, but the article does not provide solutions. 

We thank the reviewer for the input. We have now extended the discussion of some of the results, exploiting more insights on stigmatization and responsibility (as suggested). We also added, in the same section, further suggestions for experts/professionals, expressing our take on a solution to deal with prejudices and personal opinion in those settings.

7. The bibliography does not present major errors but there are some small flaws to improve, for example the journal in italics, etc.

We checked the bibliography, correcting typos and changing the format into APA style (as indicated by the journal via e-mail).

Reviewer 2 Report

Comments and Suggestions for Authors

This qualitative study explored how victims and offenders of gender violence described their roles and emotions from three perspectives, namely, victims, offenders, and clinic experts. The results indicate that both victims and offenders described experiences of gender violence as fearful and self-guilt, whereas offenders often exhibited anger and pathology. Based in part on their findings, the authors proposed a framework for clinical interventions to reduce the psychological impact of gender violence on victims and modify offenders’ violent behaviors. Below are my observations and suggestions and hope they will be useful for authors’ revision endeavors.

First, the authors used gender violence and intimate partner violence (IPV) separately and interchangeably throughout the manuscript but never defined what IPV entails. Are they conceptually and empirically equivalent? I urge the authors to review Michael Johnson’s work on IPV and different forms of IPV.

Second, what is the guiding theoretical framework in this study? Was it explicitly stated?

Third, is it more appropriate to state research questions than hypotheses in qualitative studies?

Fourth, the study was conducted in Italy. It would be nice for the authors to develop a paragraph contextualizing gender violence in Italy, especially, the prevalence and consequences of gender violence. Your readers unfamiliar with Italy will welcome such social and cultural contexts of IPV.

Fifth, the sophisticated research methods are impressive, but the results are often under-interpreted. For example, the authors reported a series of chi-square values in Tables 3 and 4, what do they mean? Why are they statistically insignificant? Similarly, the results displayed in Table 5 are either statistically significant or insignificant. Why? What do they signify?

Sixth, the results that surfaced from the authors’ analyses are not new. Despite the fancy methods, the findings are consistent with a large body of empirical research conducted in the West. So, what is new here? I am not convinced by the authors’ justifications provided in the Introduction.

Finally, thorough line editing by a native speaker is highly recommended. Some of the sentences are extremely awkward.

Comments on the Quality of English Language

See above.

Author Response

First of all, we thank the reviewer for all the valuable suggestions and inputs provided. Here follows a point-by-point response to the comments:

1. First, the authors used gender violence and intimate partner violence (IPV) separately and interchangeably throughout the manuscript but never defined what IPV entails. Are they conceptually and empirically equivalent? I urge the authors to review Michael Johnson’s work on IPV and different forms of IPV.

We thank the reviewer for pointing this out. We have now added a definition of IPV, and specified in the objective of our research that our focus lies in IPV. However, we also specify (and have done the same in the text) that, accordingly with our theoretical assumptions, we apply a discursive definition of gender-based violence that subsumes also IPV situations.

2. Second, what is the guiding theoretical framework in this study? Was it explicitly stated?

We added a new short paragraph at the start of Materials and Methods reporting our theoretical references and assumptions.

3. Third, is it more appropriate to state research questions than hypotheses in qualitative studies?

The latter is indeed more appropriate, thus we added the research question in the Introduction. We chose to maintain the hypothesis as possible answers to the research question, later discussed in the related section.

4. Fourth, the study was conducted in Italy. It would be nice for the authors to develop a paragraph contextualizing gender violence in Italy, especially, the prevalence and consequences of gender violence. Your readers unfamiliar with Italy will welcome such social and cultural contexts of IPV.

We definitely agree and have added, at the end of the Introduction, a short reporting data specifically related to the Italian context.

5. Fifth, the sophisticated research methods are impressive, but the results are often under-interpreted. For example, the authors reported a series of chi-square values in Tables 3 and 4, what do they mean? Why are they statistically insignificant? Similarly, the results displayed in Table 5 are either statistically significant or insignificant. Why? What do they signify?

We thank the reviewer for pointing this out. First of all, we specified (both in Materials and Methods and Results sections) why we decided to add some statistically non-significant forms and what these latter means overall. Moreover, both for forms in Tab. 3-4 and Tab. 5 we added further details, in particular for the terms that IRaMuTeQ’s analysis pointed out as being statistically non-relevant.

6. Sixth, the results that surfaced from the authors’ analyses are not new. Despite the fancy methods, the findings are consistent with a large body of empirical research conducted in the West. So, what is new here? I am not convinced by the authors’ justifications provided in the Introduction.

We have now added some more elements of discussion based on the results we obtained, delving into the answer examples and exploiting them on stigmatization and responsibility issues. We agree that our findings are consistent with other field studies’ results, especially when looking at contents and implications. Thus, we expanded (mainly in the introduction, but also in the conclusions) the peculiarity of our research method, which provides data on the specific discursive modalities adopted by participants. This is a level of analysis that we did not find in the current literature, that we believe could lead to further insights (and maybe also different results) with larger research (considering the limitation of our study in the number of participants involved).

7. Finally, thorough line editing by a native speaker is highly recommended. Some of the sentences are extremely awkward.

With the changes made, we also checked the English formulations and adjusted the sentences that we agree were complex/difficult to understand.

Round 2

Reviewer 2 Report

Comments and Suggestions for Authors

The authors are very responsive to the reviewers' comments and suggestions and made some necessary changes. As a result, the manuscript is improved. However, at times the data analysis and interpretations are still difficult for me to follow. As I stated previously, I just didn't learn anything new. As a matter of fact, the detailed data analyses lost me in terms of what the authors wanted to achieve.

In the conclusion section, the authors stated that gender-based violence is socially constructed. But how? It was surprising that the authors didn't elaborate on it. For example, why is it socially constructed from the victims' perspectives? Offenders' perspective? And experts' perspectives? I thought this would be a perfect occasion for the authors to make sense of the research findings in terms of gender stereotypes, power dynamics, cultural norms, and so on.

Comments on the Quality of English Language

Another round of line editing is recommended.

Author Response

First of all, we would like to thank the reviewer for recognizing our effort and for providing these further suggestions. Here follows our responses to the comments:

1. However, at times the data analysis and interpretations are still difficult for me to follow. As I stated previously, I just didn't learn anything new. As a matter of fact, the detailed data analyses lost me in terms of what the authors wanted to achieve.

In order to make data interpretation more clear, we added some more reasoning in the Discussion section to explain our observations (these additions are also linked to the topic of the social construction of the roles of victim and offender). We also tried to describe in more detail the peculiarity of our methodology and what we believe is the innovative take and information it provides (its focus on the structure of participants’ narratives in Discursive Repertories and their measurement through dW).

2. In the conclusion section, the authors stated that gender-based violence is socially constructed. But how? It was surprising that the authors didn't elaborate on it. For example, why is it socially constructed from the victims' perspectives? Offenders' perspective? And experts' perspectives? I thought this would be a perfect occasion for the authors to make sense of the research findings in terms of gender stereotypes, power dynamics, cultural norms, and so on.

We understand this issue, and have further explored and elaborated on the topic of social construction (both in the Discussion and Conclusion sections) to highlight how the mutual narrative of victims, offenders and expert aligns and are linked to current social representations of them. We referred back to some already mentioned references and added some more.

3. Another round of line editing is recommended.

We checked again for typos and/or overly articulated sentences, adjusting where necessary.